# Oral step-down vs full-course intravenous antibiotic therapy for infective endocarditis: Protocol for a systematic review and meta-analysis

Jose D. Cruz-Cuevas[1,2]*, Julián Andrés Amador Bedoya[1,2]☉,
María Paula Méndez Gaitán[1,3]☉, María Camila Ramírez[1,2], Javier González Quiroga[1,2],
Adelaida Rodríguez Villegas[1,2], Antonia Sanín Reyes[1,4]

1 Fundación Cardioinfantil – Instituto de Cardiología, Bogotá, Colombia, 2 School of Medicine and Health Sciences, Universidad del Rosario, Bogotá, Colombia, 3 School of Medicine, Universidad El Bosque, Bogotá, Colombia, 4 School of Medicine, Universidad de La Sabana, Bogotá, Colombia

☉ These authors contributed equally to this work.
* jcruz@lacardio.org

## Abstract

### Background

Infective endocarditis (IE) is a life-threatening condition traditionally managed with prolonged intravenous (IV) antibiotic therapy. However, sequential oral antibiotic therapy has emerged as a promising alternative in selected patients, potentially reducing hospital stay, adverse events, and healthcare costs. While the POET trial demonstrated non-inferiority of oral step-down therapy in a narrow subset of patients with left-sided IE, broader applicability remains uncertain. Observational studies and recent meta-analyses have yielded heterogeneous and inconclusive results, under-scoring the need for a comprehensive synthesis of current evidence.

### Methods

This protocol outlines the methodology for a systematic review and meta-analysis comparing the effectiveness and safety of sequential oral antibiotic therapy versus full-course IV therapy in adult patients with bacterial IE. We will include randomized controlled trials and observational comparative studies enrolling adults diagnosed with IE by Duke criteria, clinical judgment, or histopathological confirmation. The primary outcomes are all-cause mortality and clinical cure; secondary outcomes include relapse, unplanned cardiac surgery, embolic events, hospital length of stay, and adverse events. Searches will be conducted in MEDLINE, Embase, CENTRAL, and LILACS, with no restrictions on publication date or language. Risk of bias will be assessed using RoB 2 and ROBINS-I tools. Data synthesis will follow Cochrane guidelines, and certainty of evidence will be evaluated using the GRADE approach.

**Data availability statement:** No datasets were generated or analysed during the current study. All relevant data from this study will be made available upon study completion.

**Funding:** The author(s) received no specific funding for this work.

**Competing interests:** The authors have declared that no competing interests exist.

## Discussion

This review will provide a rigorous and up-to-date synthesis of the evidence on oral step-down therapy in IE, addressing populations underrepresented in prior trials. The findings will inform clinical decision-making, guideline development, and future research on optimizing antibiotic strategies for IE.

## Systematic review registration

PROSPERO registration number: CRD420251104423

## Introduction

Infective endocarditis (IE) is a serious and potentially life-threatening infection of the endocardial surface of the heart, most frequently involving native or prosthetic valves. Its clinical presentation, extent of cardiac involvement, and prognosis are highly heterogeneous, depending on factors such as the affected valve (native vs. prosthetic), anatomical location (left- vs. right-sided), causative microorganism, and host characteristics. Despite advances in antimicrobial therapy and cardiac surgery, in-hospital mortality remains high, ranging from 15% to 30% [1,2]. Left-sided IE, affecting the mitral and aortic valves, predominates and is associated with systemic complications such as embolic events, heart failure, and perivalvular abscesses, particularly in patients with structural heart disease or prosthetic valves. In contrast, right-sided IE is more commonly observed among intravenous (IV) drug users and patients with intravascular devices, typically leading to septic pulmonary emboli and a comparatively better prognosis [1,2]. The most common causative organisms include *Staphylococcus aureus*, viridans group streptococci, and *Enterococcus* species, while gram-negative and fungal etiologies account for a minority of cases [1,2]. Guidelines suggest current standard therapy requires 4–6 weeks of IV antibiotics, necessitating prolonged hospitalization or outpatient parenteral antibiotic therapy (OPAT), both of which carry significant costs, risks, and implications for patient quality of life. These challenges are particularly pronounced in low- and middle-income countries, where sustained inpatient care and structured outpatient follow-up may be limited or even nonexistent [2].

Sequential oral antibiotic therapy has emerged as a promising alternative to prolonged IV regimens in selected patients with IE. This approach involves an early switch from IV antibiotics to highly bioavailable oral agents, provided that the pathogen has been identified, susceptibility confirmed, and the patient has achieved clinical stability. The pharmacological rationale lies in the ability of oral antibiotics to sustain adequate plasma and tissue concentrations to ensure bacterial eradication, while reducing the risks associated with long-term IV access, including catheter-related infections and thrombosis. Moreover, oral therapy facilitates earlier mobilization and hospital discharge, potentially improving recovery, reducing hospital-related adverse events, and lowering healthcare costs [3].

The landmark Partial Oral Treatment of Endocarditis (POET) trial, published in 2019, was the first multicenter randomized controlled trial to demonstrate that, in

carefully selected patients with left-sided IE, switching to oral antibiotics after a minimum of 10 days of IV therapy was non-inferior to continuing a full course of IV treatment [3]. Rates of all-cause mortality, unplanned cardiac surgery, embolic events, and relapse did not differ significantly between groups. However, only about 20% of screened patients qualified, as the trial excluded those with prosthetic valves, right-sided IE, resistant pathogens, IV drug use, or immunosuppression—limiting generalizability [3,6].

Since the publishing of the POET trial, subsequent observational studies and systematic reviews have explored broader populations. However, due to the heterogeneity of the results evidence is unreliable and even biased. Rezar et al. (2020) suggested non-inferiority of oral therapy in non-critically ill patients, though the included studies were largely observational and methodologically limited [4]. More recently, Mourad et al. (2025) found no significant difference in treatment failure between oral and IV strategies in patients with Staphylococcus aureus bacteremia and IE, though rates of adverse events varied across cohorts [5]. These findings have influenced international guidelines, including the 2023 European Society of Cardiology (ESC) recommendations [6], and have fueled growing interest in oral step-down strategies. Nevertheless, the evidence remains fragmented and insufficient to justify broad implementation across all patient groups.

The primary objective of adopting a partial oral step-down strategy in IE is non-inferiority in clinical effectiveness compared with continued full-course IV therapy, rather than superiority. If comparable efficacy is achieved, oral transition offers important secondary advantages that are clinically and system-relevant, including lower rates of catheter-related complications, shorter hospital stays, reduced healthcare costs, and improved patient quality of life.

Therefore, a comprehensive and up-to-date systematic review and meta-analysis is warranted to synthesize available evidence on oral step-down antibiotic therapy in IE. By incorporating data from randomized controlled trials and observational studies, and conducting detailed subgroup analyses, this review aims to evaluate the overall effectiveness, safety, and feasibility of oral therapy compared with standard IV regimens. The results will provide essential evidence to guide clinical practice, optimize patient management, and inform future updates to international treatment guidelines [6].

## Materials and methods

### Study design

This protocol is registered in the PROSPERO database (International Prospective Register of Systematic Reviews) under registration number CRD420251104423, and is reported in accordance with the PRISMA-P guidelines (Preferred Reporting Items for Systematic Review and Meta-Analyses Protocol) [7], as detailed in S1 Table. The present manuscript outlines the methodological strategies to be applied, including search strategies, eligibility criteria, data extraction procedures, variables of interest, data synthesis methods, and the approach to managing heterogeneity. The purpose of this protocol is to ensure transparency, reproducibility, and clarity in the conduct of this systematic review and meta-analysis.

At the time of submission, the search strategy has been completed and record screening is ongoing, expected to finish by December 2025. Data extraction will begin in January 2026 and conclude by February 2026. Statistical analysis and synthesis of results are anticipated by April 2026. No results have been generated yet, and data extraction has not started.

The review will be conducted following the recommendations of the Cochrane Handbook for Systematic Reviews of Interventions [8], ensuring a rigorous and reproducible methodological approach to the synthesis of evidence on the effectiveness and safety of oral step-down antibiotic therapy compared to full-course IV therapy in adult patients with bacterial IE.

### Eligibility criteria

We will include studies that enroll adult patients aged 18 years or older with a diagnosis of bacterial IE. This diagnosis may be established through fulfillment of Duke or modified Duke criteria—whether classified as definite or possible endocarditis—or through a clinical diagnosis made by the treating physician or a multidisciplinary team, supported by

microbiological evidence even if the full Duke criteria are not met. Additionally, studies will be eligible if they report intra-operative or surgical findings consistent with endocarditis, confirmed by microbiological or histopathological analysis, or if histopathological evidence of infection is found on excised cardiac tissue or valves. Both left-sided and right-sided endo-carditis will be considered, as well as infections involving native valves, prosthetic valves, or cardiac implantable electronic devices.

The intervention of interest is sequential oral antibiotic therapy, administered after an initial period of IV treatment of any duration. This may involve oral monotherapy or combination therapy, provided that the switch occurs following IV adminis-tration. The comparator is standard full-course IV therapy, typically administered for four to six weeks without transition to oral agents.

Primary outcomes will include all-cause mortality and clinical cure, defined as resolution of signs and symptoms of infection without relapse. Secondary outcomes will include reinfection or relapse, unplanned cardiac surgery, embolic events, hospital length of stay, and adverse events such as allergic reactions, acute kidney injury, intensive care unit requirement, and catheter-related complications.

Eligible study designs will include randomized controlled trials and observational comparative studies, such as prospec-tive or retrospective cohort studies and case-control studies. We will exclude case reports, case series, narrative reviews, and editorials. No restrictions will be applied regarding publication language or year. Both peer-reviewed articles and conference abstracts will be considered eligible, provided they contain sufficient methodological detail and outcome data. Full inclusion and exclusion criteria are presented in Table 1.

## Search strategy

The bibliographic databases selected for this review are MEDLINE, Embase, CENTRAL (Cochrane Central Register of Controlled Trials), and LILACS (Latin American and Caribbean Health Sciences Literature). In addition to these elec-tronic sources, other studies will be identified through reference list checking and citation tracking of included articles

**Table 1. Eligibility Criteria. Inclusion and exclusion criteria applied for study selection.**

| Category | Inclusion | Exclusion |
|---|---|---|
| Population | Adult patients (≥18 years) with definite or possible infective endocarditis diagnosed by Duke or modified Duke criteria, clinical diagnosis with microbiological support, intraoperative/surgical confirmation, or histopathological evidence. Includes left- and right-sided IE, native and prosthetic valves, and cardiac implant-able electronic devices. | Pediatric populations; studies without sufficient diagnostic confirmation of IE; nonbacterial or noninfective causes |
| Intervention | Sequential oral antibiotic therapy (partial oral treatment), after an initial period of intravenous therapy of any duration, using high-bioavailability oral agents active against the causative pathogens. | Studies without oral antibiotic therapy; studies where oral ther-apy is used only for noncardiac infections. |
| Comparator | Full-course intravenous antibiotic therapy, typically 4–6 weeks in duration. | Studies without a comparator group. |
| Outcomes | Primary: all-cause mortality, clinical cure. Secondary: relapse or reinfection, unplanned cardiac surgery, embolic events, hospital length of stay, adverse events (allergic reactions, acute kidney injury, ICU requirement, catheter-related complications). | Studies not reporting clinical out-comes of interest. |
| Study type | Randomized controlled trials, prospective and retrospective cohort studies, and case-control studies. | Case reports, case series, reviews, editorials, noncompara-tive studies. |

("snowballing"). No restrictions will be applied regarding publication date or language. We will include studies published from database inception through 31 December 2025. Studies published in any language will be considered eligible. When potentially relevant studies are identified in languages other than English or Spanish, translation will be performed using automated tools or professional assistance to ensure accurate screening and data extraction. This strategy aims to reduce language bias and enhance the comprehensiveness of the review.

The primary search strategy combines controlled vocabulary and free-text terms related to oral antibiotic therapy and IE. The core search string is: ("PO antibiotics" OR "oral drug administration" OR "sequential therapy" OR "de-escalation" OR "switch therapy" OR "oral antibiotics" OR ("Administration, Oral"[MeSH] AND "Anti-Bacterial Agents"[MeSH]) OR "oral therapy" OR "step-down therapy") AND endocardit*. This strategy will be adapted for each database to optimize sensitivity and specificity. The full search strategies for all databases are provided in the supporting information (S2 Table).

## Screening procedure

All references retrieved from the database searches will be imported into the Rayyan web application [9], where duplicate records will be identified and removed prior to screening. Two reviewers will independently assess the titles and abstracts of all retrieved records to determine their potential eligibility. During full-text screening, the same pair of reviewers will independently assess each article in detail, applying the eligibility criteria consistently. Discrepancies will be resolved through discussion and consensus, and if necessary, a third reviewer will adjudicate (JDC).

In cases where the full text of a study cannot be retrieved, corresponding authors or publishers will be contacted to obtain additional information. If sufficient data cannot be obtained, the study will be excluded, and the reasons for exclusion will be documented. The study selection process will be summarized in a PRISMA flow diagram, indicating the number of records identified, screened, excluded, and ultimately included in the review.

## Data extraction

Data from eligible studies will be extracted independently by two reviewers using a standardized data collection form developed in REDCap [10,11]. The extracted information will include study-level characteristics such as first author, year of publication, country, study design, sample size, and duration of follow-up. Population-level data will include age, sex distribution, comorbidities, type and location of valve involvement (e.g., native or prosthetic, left- or right-sided), and causative microorganisms.

Details of the intervention will include the duration of initial IV therapy, oral antibiotic regimens used (monotherapy or combination), total treatment duration, and criteria for switching to oral therapy. Comparator data will include the IV antibiotics administered and the duration of IV therapy. Outcomes to be extracted will include all-cause mortality, clinical cure, relapse or reinfection, unplanned cardiac surgery, embolic events, hospital length of stay, and adverse events such as allergic reactions, acute kidney injury, ICU requirement, and catheter-related complications.

Discrepancies in data extraction will be resolved through discussion between reviewers, and if necessary, by consultation with a third reviewer. When relevant data are missing or unclear, attempts will be made to contact the corresponding authors of the original studies to obtain clarification or additional information.

## Risk of bias assessment

The risk of bias of the included studies will be assessed independently by two reviewers. For randomized controlled trials, the revised Cochrane risk-of-bias tool for randomized trials (RoB 2) will be applied [12]. This tool evaluates five domains: bias arising from the randomization process, bias due to deviations from intended interventions, bias due to missing outcome data, bias in measurement of the outcome, and bias in selection of the reported result.

For non-randomized comparative studies, the ROBINS-I tool will be used to assess potential biases across seven domains: bias due to confounding, bias in the selection of participants into the study, bias in classification of interventions, bias due to deviations from intended interventions, bias due to missing data, bias in measurement of outcomes, and bias in selection of the reported result [13].

Any disagreements between reviewers will be resolved through discussion, and if consensus cannot be reached, a third reviewer will arbitrate. When relevant information is missing or unclear, study authors will be contacted to obtain clarification. The overall risk-of-bias judgments will be reported in both tabular and graphical formats to facilitate interpretation and transparency.

## Data synthesis and statistical analysis

We will conduct a quantitative synthesis of the included studies if sufficient homogeneity is observed in terms of study design, population characteristics, interventions, and outcomes. For dichotomous outcomes, risk ratios (RR) with 95% confidence intervals will be calculated. For continuous outcomes, mean differences (MD) or standardized mean differences (SMD) with 95% confidence intervals will be used, depending on the measurement scales and variability across studies.

When studies report medians and ranges or interquartile ranges instead of means and standard deviations, we will apply the methods proposed by Hozo et al. [14] and Wan et al. [15] to estimate the corresponding parameters. We anticipate important clinical and methodological heterogeneity across studies; therefore, meta-analyses will be performed using a random-effects model based on the DerSimonian and Laird method. Statistical heterogeneity will be assessed using the $I^2$ statistic and the Chi-square test, with $I^2$ values greater than 50% considered indicative of substantial heterogeneity.

We acknowledge the marked clinical heterogeneity of IE. As a biologically and clinically informed a priori hypothesis, we consider that the feasibility and safety of oral step-down therapy may be greater in relatively less complex scenarios—for example, native valve endocarditis, left-sided disease without extensive local complications, infections caused by lower-virulence microorganisms, and patients who achieve early clinical stability without heart failure, persistent bacteremia, or embolic phenomena.

This hypothesis will not be used to restrict eligibility; instead, it will be examined through prespecified subgroup and sensitivity analyses, including type of valve involved (native vs. prosthetic), causative microorganisms, presence of prosthetic material or cardiac devices, duration of IV therapy before switching, and specific oral antibiotic regimens. Also, methodological heterogeneity might be present due to wide eligibility criteria accounting for scarce evidence in this topic. To evaluate the contribution of methodological differences to heterogenity, subgroups analysis will include study design (RCT vs observational) and direction (prospective vs retrospective). Sensitivity analyses will be performed to explore the robustness of the findings, including exclusion of studies at high risk of bias and assessment of the impact of missing data.

To evaluate potential publication bias, we will examine funnel plot asymmetry when at least ten studies are available for a given outcome. Additionally, we will discuss the nature and extent of missing data and reporting limitations, and contact study authors when necessary to clarify or obtain unpublished information. No data imputation will be performed.

If meta-analysis is not feasible due to substantial heterogeneity, limited number of studies, or insufficiently comparable data, a structured narrative synthesis will be provided, including summary tables and descriptive text.

All statistical analyses will be performed using R software, employing the meta, metafor, and dmetar packages for meta-analysis and visualization.

## Certainty of evidence

The certainty of the evidence for each primary and secondary outcome will be assessed using the Grading of Recommendations, Assessment, Development, and Evaluation (GRADE) approach. This evaluation will be conducted independently by two reviewers, considering the following five domains: risk of bias, inconsistency, indirectness, imprecision, and

publication bias. For each outcome, the certainty of the evidence will be rated as high, moderate, low, or very low, based on the overall assessment across these domains.

Findings will be summarized in Summary of Findings (SoF) tables, generated using the GRADEpro GDT software, to provide a transparent and structured presentation of the quality of evidence and the magnitude of effect for each outcome.

## Discussion

This systematic review and meta-analysis will provide a comprehensive synthesis of the current evidence regarding the efficacy and safety of sequential oral antibiotic therapy compared with standard IV treatment in patients with IE. The findings are expected to clarify whether oral step-down therapy yields comparable outcomes in terms of mortality, clinical cure, relapse, need for cardiac surgery, and embolic complications, while potentially reducing hospital length of stay and adverse events associated with prolonged IV access.

Importantly, the experimental strategy—switching to oral antibiotics—is not intended to demonstrate superiority over IV therapy, but rather non-inferiority. If clinical outcomes are similar, oral therapy may represent a favorable alternative due to its advantages in terms of patient comfort, reduced adverse events, lower costs, and decreased resource utilization. However, establishing a formal non-inferiority margin is challenging in the context of a meta-analysis, as such thresholds are typically defined prospectively in randomized trials and may vary depending on the outcome assessed and the clinical setting in which the strategy is implemented.

Although this review does not define a specific non-inferiority margin a priori, the point estimates and confidence intervals derived from the included studies may still be informative. These results can be interpreted in light of thresholds used in previous trials or contextualized according to clinical judgment and local implementation priorities. This interpretative flexibility is particularly relevant in settings with limited resources, where the trade-offs between efficacy and feasibility may differ from those in high-income contexts.

By including both randomized controlled trials and high-quality observational studies, this review will expand upon the findings of the POET trial and subsequent investigations, addressing populations that were previously underrepresented—such as patients with prosthetic valves, right-sided endocarditis, IV drug use, or infections caused by resistant organisms. Furthermore, subgroup and sensitivity analyses may help delineate which patient groups are most likely to benefit from this strategy, as well as identify the optimal timing and choice of oral antibiotic regimens.

By preregistering methods in PROSPERO (CRD420251104423) and adhering to PRISMA-P/Cochrane guidance, this protocol improves transparency, reduces analytic flexibility, and supports reproducibility prior to knowing results. This is especially pertinent for a heterogeneous clinical question—oral step-down in IE—where planned subgroups and sensitivity analyses must be specified a priori and any deviations reported transparently.

Ultimately, the evidence generated by this review will support clinical decision-making, inform future updates to international guidelines, and highlight areas where further research is needed to strengthen the evidence base for the management of IE.

## Supporting information

**S1 Table PRISMA-P 2015 checklist.**
(DOCX)

**S2 Table Search Strategy.**
(DOCX)

## Acknowledgments

We thank the Fundación Cardioinfantil – Instituto de Cardiología for institutional support.

## Author contributions

**Conceptualization:** Jose D. Cruz-Cuevas.

**Investigation:** Julián Andrés Amador Bedoya, María Camila Ramírez, Javier González Quiroga, Adelaida Rodríguez Villegas, Antonia Sanín Reyes.

**Methodology:** Jose D. Cruz-Cuevas, Maria Paula Méndez Gaitán.

**Project administration:** Julián Andrés Amador Bedoya.

**Resources:** Antonia Sanín Reyes.

**Writing – original draft:** Jose D. Cruz-Cuevas.

**Writing – review & editing:** Julián Andrés Amador Bedoya, Maria Paula Méndez Gaitán, María Camila Ramírez, Javier González Quiroga, Adelaida Rodríguez Villegas, Antonia Sanín Reyes.

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
