## [Decision Letter · Decision Letter 0]

26 Feb 2026

Dear Dr. Cruz Cuevas,

We look forward to receiving your revised manuscript.

Kind regards,

Robert Jeenchen Chen, MD, MPH, ChFC®, EA

Academic Editor

PLOS One

Journal Requirements:

Reviewers' comments:

Reviewer's Responses to Questions

**Comments to the Author**

1. Does the manuscript provide a valid rationale for the proposed study, with clearly identified and justified research questions?

Reviewer #1: Yes

Reviewer #2: Partly

2. Is the protocol technically sound and planned in a manner that will lead to a meaningful outcome and allow testing the stated hypotheses?

Reviewer #1: Yes

Reviewer #2: Partly

3. Is the methodology feasible and described in sufficient detail to allow the work to be replicable?

Reviewer #1: Yes

Reviewer #2: No

4. Have the authors described where all data underlying the findings will be made available when the study is complete?

The PLOS Data policy requires authors to make all data underlying the findings described in their manuscript fully available without restriction, with rare exception, at the time of publication. The data should be provided as part of the manuscript or its supporting information, or deposited to a public repository. For example, in addition to summary statistics, the data points behind means, medians and variance measures should be available. If there are restrictions on publicly sharing data—e.g. participant privacy or use of data from a third party—those must be specified.requires authors to make all data underlying the findings described in their manuscript fully available without restriction, with rare exception, at the time of publication. The data should be provided as part of the manuscript or its supporting information, or deposited to a public repository. For example, in addition to summary statistics, the data points behind means, medians and variance measures should be available. If there are restrictions on publicly sharing data—e.g. participant privacy or use of data from a third party—those must be specified.

Reviewer #1: Yes

Reviewer #2: No

5. Is the manuscript presented in an intelligible fashion and written in standard English?

Reviewer #1: Yes

Reviewer #2: Yes

You may also provide optional suggestions and comments to authors that they might find helpful in planning their study.

Reviewer #1: Infective endocarditis (IE) is typically treated with prolonged intravenous (IV) antibiotics. Sequential oral antibiotic therapy has emerged as a potential alternative for selected patients, with possible benefits such as reduced hospital stay, adverse events, and costs. While the POET trial demonstrated non‑inferiority of oral step‑down therapy in a limited group with left‑sided IE, its broader applicability remains unclear. Evidence from observational studies and meta‑analyses is inconsistent, warranting a comprehensive evaluation. This review synthesizes current evidence on oral step‑down therapy to guide clinical practice and future research.

Reviewer #2: We have some comments to be taken into consideration by the authors:

1- The authors should elaborate on a better rationale. (e.g., does the shift from intravenous to oral expected to give better out come?!, what categories and classes of infective endocarditis could derive the greatest benefit with oral regime?!.

2- In the methodology, the authors should highlight the years of enrollment of studies. (e.g., including the studies published from Jan 2008 to Jan 2026).

3- It should be clear in the methodology, how the authors would manage heterogeneity between the studies. Heterogeneity between retrospective and prospective study, between trial design and off label studies, between native valve endocarditis cohort and prosthetic valve endocarditis cohort.

4- Discuss in a better manner, what is the added value of publishing a protocol for systematic review?!.

Regards

.

Reviewer #1: No

Reviewer #2: **Yes:**Rami-Riziq-Yousef-AbumuaileqRami-Riziq-Yousef-AbumuaileqRami-Riziq-Yousef-AbumuaileqRami-Riziq-Yousef-Abumuaileq

---

## [Author Response · Author response to Decision Letter 1]

30 Mar 2026

We thank the Editor and the Reviewers for their careful evaluation of our protocol and for their constructive comments, which have helped us improve the clarity, rigor, and transparency of the manuscript. Below, we provide a detailed, point by point response. All suggested changes have been incorporated into the revised version of the manuscript, and the relevant sections are explicitly indicated.

---

## [Decision Letter · Decision Letter 1]

12 Apr 2026

Oral step-down vs full-course intravenous antibiotic therapy for infective endocarditis: Protocol for a systematic review and meta-analysis

PONE-D-25-59799R1

Dear Dr. Cruz Cuevas,

We’re pleased to inform you that your manuscript has been judged scientifically suitable for publication and will be formally accepted for publication once it meets all outstanding technical requirements.

Kind regards,

Robert Jeenchen Chen, MD, MPH, ChFC®, EA

Academic Editor

PLOS One

Additional Editor Comments (optional):

Reviewers' comments:

Reviewer's Responses to Questions

**Comments to the Author**

1. Does the manuscript provide a valid rationale for the proposed study, with clearly identified and justified research questions?

Reviewer #2: Yes

2. Is the protocol technically sound and planned in a manner that will lead to a meaningful outcome and allow testing the stated hypotheses?

Reviewer #2: Partly

3. Is the methodology feasible and described in sufficient detail to allow the work to be replicable?

Reviewer #2: Yes

4. Have the authors described where all data underlying the findings will be made available when the study is complete?

The PLOS Data policy requires authors to make all data underlying the findings described in their manuscript fully available without restriction, with rare exception, at the time of publication. The data should be provided as part of the manuscript or its supporting information, or deposited to a public repository. For example, in addition to summary statistics, the data points behind means, medians and variance measures should be available. If there are restrictions on publicly sharing data—e.g. participant privacy or use of data from a third party—those must be specified.requires authors to make all data underlying the findings described in their manuscript fully available without restriction, with rare exception, at the time of publication. The data should be provided as part of the manuscript or its supporting information, or deposited to a public repository. For example, in addition to summary statistics, the data points behind means, medians and variance measures should be available. If there are restrictions on publicly sharing data—e.g. participant privacy or use of data from a third party—those must be specified.

Reviewer #2: No

5. Is the manuscript presented in an intelligible fashion and written in standard English?

Reviewer #2: Yes

You may also provide optional suggestions and comments to authors that they might find helpful in planning their study.

Reviewer #2: The authors have addressed properly our comments and the manuscript has been improved.

Kindest regards

.

Reviewer #2: **Yes:**Rami Riziq Yousef AbumuaileqRami Riziq Yousef AbumuaileqRami Riziq Yousef AbumuaileqRami Riziq Yousef Abumuaileq

---

## [Editor Report · Acceptance letter]

PONE-D-25-59799R1

PLOS One

Dear Dr. Cruz-Cuevas,

I'm pleased to inform you that your manuscript has been deemed suitable for publication in PLOS One. Congratulations! Your manuscript is now being handed over to our production team.

Kind regards,

on behalf of

Dr. Robert Jeenchen Chen

Academic Editor

PLOS One